# Surface-Enhanced Raman Sensing of Semi-Volatile Organic Compounds by Plasmonic Nanostructures

**DOI:** 10.3390/nano11102619

**Published:** 2021-10-05

**Authors:** Nguyễn Hoàng Ly, Sang Jun Son, Soonmin Jang, Cheolmin Lee, Jung Il Lee, Sang-Woo Joo

**Affiliations:** 1Department of Chemistry, Gachon University, Seongnam 13120, Korea; nguyenhoangly2007@gmail.com; 2Department of Chemistry, Sejong University, Seoul 05006, Korea; sjang@sejong.edu; 3Department of Chemical & Biological Engineering, Seokyeong University, Seoul 02713, Korea; cheolmin@skuniv.ac.kr; 4Korea Testing & Research Institute, Gwacheon 13810, Korea; 5Department of Chemistry, Soongsil University, Seoul 06978, Korea

**Keywords:** semi-volatile organic compounds, surface-enhanced Raman scattering, plasmonic resonance, Raman spectroscopy, noble metal nanostructures

## Abstract

Facile detection of indoor semi-volatile organic compounds (SVOCs) is a critical issue to raise an increasing concern to current researchers, since their emissions have impacted the health of humans, who spend much of their time indoors after the recent incessant COVID-19 pandemic outbreaks. Plasmonic nanomaterial platforms can utilize an electromagnetic field to induce significant Raman signal enhancements of vibrational spectra of pollutant molecules from localized hotspots. Surface-enhanced Raman scattering (SERS) sensing based on functional plasmonic nanostructures has currently emerged as a powerful analytical technique, which is widely adopted for the ultra-sensitive detection of SVOC molecules, including phthalates and polycyclic aromatic hydrocarbons (PAHs) from household chemicals in indoor environments. This concise topical review gives updated recent developments and trends in optical sensors of surface plasmon resonance (SPR) and SERS for effective sensing of SVOCs by functionalization of noble metal nanostructures. Specific features of plasmonic nanomaterials utilized in sensors are evaluated comparatively, including their various sizes and shapes. Novel aptasensors-assisted SERS technology and its potential application are also introduced for selective sensing. The current challenges and perspectives on SERS-based optical sensors using plasmonic nanomaterial platforms and aptasensors are discussed for applying indoor SVOC detection.

## 1. Introduction

Functional plasmonic nanostructures have gained increasing interest in energy and environmental fields [1]. SERS has recently been introduced for effective sensing combined with noble metal systems by introducing hotspots to enhance electromagnetic enhancements [2,3]. Biosensing [4,5] can also be achieved using novel nanostructures [6]. Semi-volatile organic compounds (SVOCs) have a range of boiling points, from 260 to 400 °C as a sub-group of volatile organic compounds (VOCs) [7]. SVOCs, including plasticizers, flame retardants, per-fluorinated compounds, antioxidants, per- and polyfluoroalkyl substances, etc., have been widely used as additives in several commercial products such as textiles, polymers, cleaning products, electronic devices, plastic items, etc., which are usually used in houses [8]. SVOCs can slowly and directly be emitted from source materials to the indoor environment under various phases, such as dust fractions [9], and gas [10,11]. Subsequently, these released SVOCs can critically impact health due to their uptake via various pathways such as human skin, clothing, and hair [12,13,14]. Polybrominated diphenyl ethers [15] have been found to enhance emissions from indoor sources by contacting dust directly, causing potential health risks, whereas the transport of phthalates in indoor air and dust has been reported to correlate to allergic diseases [16,17,18]. Furthermore, the authors have conducted the studies on correlated issues such as either direct or air-mediated SVOC transfer from indoor sources to dust [9], SVOCs in dust and air [10,19], the time constant for SVOC’s dermal absorption from indoor gas [11], distribution of SVOCs in a residential environment [20], SVOC’s dermal uptake directly from the air [21] or between clothing and skin [22], controlling emissions of organophosphate flame retardants in indoor environments [23], and health ranking of indoor SVOCs [24].

Recently, many authors managed to assemble uniform two-dimensional and three-dimensional functional plasmon nanostructures for gas sensing applications [25,26,27]. The plasmonic resonance-based optical sensor is a technique that enhances the signal of analyte molecules absorbed on plasmonic material surface, relying on enhancing electromagnetic and chemical phenomena [28]. Surface plasmon-based sensors have been widely adopted in various fields, such as biology [29], biosensing [30], and environment [31]. Various kinds of optical sensors based on plasmonic resonance of nanostructures, including SPR, SERS, and metal-enhanced fluorescence, have been reported over the past decade [32]. SERS has been developed not only as a fingerprint spectral technology [33] but also as a strong analytical method that exhibits numerous detections of hazardous compounds, such as 2-naphthalene thiol [34], H_2_S [35], ricin B chain in human blood [36], VOCs [37], polychlorinated biphenyl (PCB) [38], biomolecules [39], hexachlorocyclohexane pesticides [40], organic pollutants [41], and microplastics [42]. Since SERS-based sensor methods have been exhibited as an efficient analytical tool for identifying target molecules, nanostructured material substrate usages have attracted the considerable attention from many researchers.

Nanostructured materials of noble metals have been crucial in successfully developing the SERS-based sensing method due to their plasmonic property [25,32]. The shapes and sizes of nanostructured materials are also critical in SPR phenomena [43]. Some researchers have investigated the synthesis of different sized nanospheres of noble materials and various shapes of non-spherical particles. For example, various nanostructured materials have been discovered, including AuNPs [44], Ag nanocubes [45], Au nanostars [46], and Ag nanoplates [47]. Both metal and non-metallic nanostructures and nanohybrids such as graphene oxide (GO)-anisotropic noble metal hybrid [38] and Ag nanoplate-deposited SiO_2_/Si wafer [48] have been developed to enhance the signal of SERS. Nanostructured pyramid of Ag–Fe-embedded GO template [49], core–shell of Au nanorods@Ag nanocubes [50], nanotextured silicon decorated with Ag–Au alloy nanoparticles [51], porous zeolite imidazole framework-wrapped urchin-like Au–Ag nanocrystals [40], Au-coated Si nanocone array [52], etc., were introduced as functional plasmonic nanostructures.

There have been reports on the hazardous effects of VOCs [53,54] and SVOCs [12,55,56] found in consumer products, indoor environments, and dust. Indoor surface physics [57] and chemistry [58] have been considered to investigate the chemical reactions on indoor surfaces as a significant key in monitoring air quality in houses, where humans spend much of their time. Although several conventional technologies of gas (or liquid) chromatography and mass spectrometry for detecting SVOCs and organic pollutants have been reported with a very low detection limit [59,60], developing a noble method of plasmonic resonance-based optical sensors of indoor SVOCs remains important. To the best of our knowledge, no reports on SERS-based sensors for indoor SVOCs exist. Hybrid nanostructures utilizing plasmonic phenomena are also included for functional nanomaterials. This topical review will discuss the recent development and trends in the design of unique nanostructured materials, aiming at efficient detection of indoor SVOCs (Figure 1). The main content used for optical detection of indoor SVOCs using plasmonic resonance-based optical sensors can be distributed into three different parts: (1) SERS detection based on nanostructures of pure noble metal materials, (2) hybrid platforms-assisted SERS sensors, and (3) aptasensor-introduced optical detection.

## 2. Pure Noble Metal Nanostructured Material-Based SERS Sensor

The most significant factor of the successful SERS sensor is the type of plasmonic nanostructured materials that have been distributed in SERS-active platforms. Among many materials, noble metals, such as Au and Ag, have emerged as potential materials that have contributed to the fabrication of several novel SERS-active platforms [25,32]. Furthermore, the sensitivity of the SERS method depends on the size and geometrical nanostructures of noble metals due to the plasmonic resonance phenomena occurring on their surface. Therefore, a series of designed strategies for pure noble metal nanostructured materials have been discovered that aim at maximizing enhancement factors, not only enhancement of electromagnetic fields but also amplification of SERS signals. For example, many studies have been recently investigated, including silver nanoparticle (AgNP)-based SERS trace detection of 16 typical polycyclic aromatic hydrocarbons [61], AuNP-based [62] and AgNP-assisted [63,64] SERS rapid determination of bisphenol A (BPA), AuNP arrays-introduced SERS sensitive detection of butyl benzyl phthalate (BBP) [65,66], and silver nanorod-based SERS detection of benzo(a)pyrene (BaP) [67].

As shown in Figure 2, a series of studies on silver nanorod-based, AgNP-based, and AuNP-based SERS methods have been published on trace detection of various indoor SVOCs as well as BaP, BBP, and polycyclic aromatic hydrocarbons. Figure 2A introduces an efficient SERS detection of BBP using ~18 nm spherical AuNPs; first, BBPs have been isolated by extracting with organic solvent as cyclohexane (CYH). Subsequently, an amount of ethanol has been added to this mixture, inducing the self-assembly of AuNPs at the interface between CYH and water. Within 10–20 s, aggregated AuNP arrays have been formed with many hotspots. After being transferred onto the surface of clean silicon wafers, self-assembled AuNP arrays containing BBPs have been checked by means of SERS. Figure 2B shows a schematic diagram of an efficient colorimetric and SERS sensing of BBP d using *β*-cyclodextrin (*β*-CD)-stabilized AuNPs. *β*-CD molecules have played significant roles of Raman probes to capture BBP. Due to the AuNP-correlated hotspots, BBP could be quantified either with the naked eye or by means of SERS. Figure 2C demonstrates a silver nanorod-based SERS method for a trace detection of BaP in real samples as well as river water and soil. Due to the unique structures of SERS substrates with highly dense hotspots, the detection limit of BaP has been estimated to be as low as 1 ppm and 10 ppm in river water and soil, respectively. Figure 2D illustrates a schematic diagram for AgNP-assisted SERS detection of 16 typical PAHs in water. Based on plasmonic AgNPs and the liquid extraction step, the SERS method has been successfully applied to a trace detection of PAHs in real water environments containing contaminants. Herein, PAHs have been isolated from water by extraction with a nonpolar organic solvent. Subsequently, three steps including separation, transfer, and volatilization have been performed. Finally, acetonitrile has been used to elute the extracted PAHs to obtain solution for measured SERS spectra.

An efficient SERS substrate has been developed by the vertical array of Ag nanoplates on the substrate’s surface [47]. Since Ag nanoplates have been well-aligned on surface platforms, inducing not only achieving gaps between near Ag nanoplates but also obtaining many sufficient hotspots on the whole substrate. This Ag nanoplate-assembled substrate exhibits strong effective plasmon resonance phenomena, which have been applied for SERS sensitive detection of 3,3′,4,4′-tetrachlorobiphenyl (PCB-77). Furthermore, in mixed solutions, this SERS substrate has successfully distinguished the characterization of Raman peaks of different PCBs. Thus, Ag nanoplate-assembled substrate based on the vertically well-aligned Ag nanoplate has been a potential SERS substrate for robust, direct, and trace detection of PCBs (Figure 3).

Moreover, either benzothiazole (BZT) or its derivatives have been used as a popular ingredient in household products due to their pharmaceutical activities and biological features. Therefore, evaluating harmful health effects of their emissions inducing exposure from sources to indoor air is important. Authors have studied the evaporation of BZT from glass surfaces by combining AuNP-based SERS sensor and gas-phase infrared spectrum [44]. BZT exposure early warning is among the major concerns of environmental researchers. This investigation has been reported on optical sensors applications for monitoring BZT directly from glass surfaces. Following the fabrication of a BZT thin film on the surface, the SERS sensors method has been performed by loading AuNPs on glass surfaces. Time-dependent SERS spectral intensities of BZT have decreased, indicating substantial evaporation of BZT from the glass surface to indoor air. Simultaneously, BZT concentration in indoor air has been estimated via infrared spectra.

## 3. Hybrid Nanostructured SERS Sensors

Hybrid nanostructured materials have also been widely introduced to fabricate various SERS substrates that are applied in many fields such as biological sensing and environmental monitoring. Hybrid nanomaterials have more advantages of providing the multiple physicochemical properties than pure noble metals. Several studies have been reported for fabricating various novel hybrid nanomaterials exhibiting not only uniform geometries but also unique plasmonic properties, which aimed at developing reproducible SERS platforms. Extensive efforts have been put on the novel and robust SERS-active platforms such as SiO_2_@Ag-based composite nanospheres [68], plasmonic core–shell Au nanospheres@Ag nanocubes for phthalate plasticizer detection [69], Au@AgNP-assisted highly sensitive detection of BPA [70], up-conversion nanoparticle-decorated AuNPs for the determination of dibutyl phthalate [71], graphene monolayer-coated AgNP-based SERS sensitive detection of BPA [72], and bimetallic plasmonic Au@Ag nanocuboid-introduced SERS detection of phthalate plasticizers [73].

As shown in Figure 4, several studies on hybrid nanomaterial-based SERS methods have been reported on a trace detection of various indoor SVOCs such as PAEs and BPA. Based on the self-oxidative-polymerization of dopamine, the bio-inspired nanostructures of molecularly imprinted polymers (MIP) have been successfully designed as a template for in situ fabrication of AuNPs [74]. The AuNP-coated MIP template has a three-dimensional (3D) nanoplatform, which has been used for selective and sensitive detection of phthalate plasticizers through SERS spectroscopy. These SERS plasmonic nanostructures have been well controlled to produce many “hotspots” on the surface of nanocomposites. Figure 4B illustrates a SERS detection of PAEs in liquid samples based on a plasmonic core–shell nanocuboid of bimetallic Au@Ag. Herein, Au@Ag nanocuboids have been successfully fabricated by a Au nanorod core and a Ag cuboid shell leading to the induction of more effective plasmonic resonance than pristine Au nanorods. Due to a unique structure of Au@Ag nanocomposites that induce strong signals, this SERS sensor indicated that detection limits of BBP and DEHP were estimated to be as low as 10^−9^ M. As shown in Figure 4C, a sensitive detection of BPA has been investigated by a solid-phase microextraction (SPME) with a combination of the SERS method. AgNPs were in situ synthesized on the surface of Si fiber leading to form AgNP-coated Si fiber. Subsequently, this structure has been modified with a monolayer of graphene resulting in the generation of a novel nanocomposite including Si fiber, AgNPs, and graphene. This unique structure exhibits not only as a SERS platform but also as an SPME fiber to detect BPA. Due to the effective cooperation of SPME and SERS, this technique shows an excellent capability of BPA detection with a detection limit as low as 1 μg/L. Figure 4D demonstrates a phthalic acid ester (PAE) detection by means of SERS and up-conversion fluorescence. Using AuNPs to decorate up-conversion nanoparticles (UCNPs), a new bimodal platform has been successfully developed in this report. In this nanocomposite, UCNPs have been employed as a fluorescent signal molecule, whereas AuNPs have been used as SERS templates with an aptamer served as a targeted detection of PAEs. This nanocomposites-assisted detection of PAEs has demonstrated the detection limits of 0.0087 and 0.0108 ng/mL for fluorescence sensors and SERS methods, respectively.

An efficient SERS-active substrate has been prepared by the well-designed synthesis of a core–shell zeolite imidazole framework (ZIF-8)-wrapped Au–Ag alloyed nanocrystals in an urchin shape [40]. This novel nanostructure exhibits high-density tips with a thickness of about 100 nm, which has been well-fabricated by adding the pre-formed plasmonic nanoparticles as Au–Ag alloyed nanocrystals into the ZIF-8 precursor. The thickness of the ZIF-8 shell layer has been well-controlled with a size of 20 nm by the hexadecyl trimethyl ammonium bromide (CTAB) concentration. This core–shell configuration of ZIF-8-wrapped urchin-inspired Au–Ag alloyed nanocrystals can be applied as a highly efficient SERS platform for trace detection of hexachlorocyclohexane (HCH) molecules. This work also shows the potential application in detecting small molecules of VOC groups using SERS platforms (Figure 5). Furthermore, an ultra-sensitive and reproducible SERS platform has been synthesized by the assembly of GO layers and anisotropic nanostructured noble metal hybrid materials, such as Au nanostars (AuNSts) and flower-inspired AgNPs [38]. These well-controlled AgNFs-GO-AuNSts nanostructures have possessed ultra-sensitive detection of PCBs at a level as low as 3.4×10^−6^ M due to the coupling effect of multi-dimensional plasmon. This substrate can be used not only to identify Raman peaks of different PCBs in their mixture but also to detect various pollutants in the environment (Figure 6).

Despite extensive efforts to improve plasmonic resonance-based optical sensors, it is continuously needed to conduct additional research to properly design several novel plasmonic nanostructured materials for various platforms using optical sensors, which aims at more sensitive and selective detection of indoor SVOCs. Understanding both plasmonic behaviors and interfaces, the authors have discovered that these new nanomaterial-based aptasensors have demonstrated their potential ability for multi-detection techniques overlapping areas of SERS, fluorescence, colorimetric, and SPR. Extensive efforts have been made toward the potential aptasensor-assisted plasmonic resonance-based efficient sensors for great advantages in sensor applications. The enhanced signals were attributed to plasmon resonance inducing high sensitivity. In the following part, this review tries to highlight several significant contributions from aptasensor-assisted and plasmonic resonance-based materials according to selected interesting examples.

## 4. Aptasensor-Introduced Optical Sensors

Recently, aptasensors with short single-stranded oligonucleotides of either ribonucleic acid (RNA) or deoxyribonucleic acid (DNA) have attracted the attention from many researchers to demonstrate their selectivity and sensitivity features. Because of high affinities, aptasensor-assisted plasmonic resonance optical sensors have been widely adopted as a powerful analytical tool in many fields of applications, such as clinical diagnostics [75], food [76], and environmental pesticide detection [77,78]. The plasmon resonance effects depend not only on the nanostructured materials but also on the configuration of aptasensors. The aptasensor-active optical sensors based on plasmonic resonance have introduced a carbon quantum dot-labeled aptamer-based fluorescent method for dibutyl phthalate [79], an optical fiber-based aptasensor-assisted plasmonic biosensor for BPA [80], asymmetric plasmonic aptasensor-based BPA [81], BPA detection using an AuNP-based colorimetric aptasensor [82], quantum dot aptasensor for di-2-ethylhexyl phthalate (DEHP) [83], localized SPR-assisted electrochemical impedance spectroscopy for DEHP [84], and AuNP-sensitized ZnO nanopencil-based detection of BPA [85].

As shown in Figure 7, several studies on the nanomaterial-assisted aptasensor-based optical sensors methods have been investigated for selective detection of various indoor SVOCs of BPA and DEHP. Herein, the authors have developed several new nanosystems to be applied for various sensor technologies, including colorimetric, fluorescence, and SPR. The colorimetric technique has been known as a normal analytic tool that has been widely adopted for use in detecting analyte molecules via comparing color changes. Fluorescence [86] and SPR [87] technologies have emerged as sensitive and selective analysis that are applied in many fields, such as biomedicine [88], bioimaging [89], and environmental detections [90]. Figure 7A shows an aptasensor-based AuNP-assisted colorimetric detection of BPA. Herein, the aptasensor has been well-designed to consist of AuNPs and a specific 24-bp aptamer of BPA. Aptamer-conjugated AuNP complex has been formed via an electrostatic interaction. This complex will induce an SPR shift in the presence of BPA. Pristine AuNPs exhibit a red color. Because of the presence of BPA leading to the aggregation of AuNPs, the color may change to blue, which can be observed with the naked eye. Based on BPA concentration-correlated blue color changes of aptamer-conjugated AuNP complex, the colorimetric detection has been successfully performed with a limit as low as 0.004 nM. Despite the presence of various BPA analogs, this method achieved a selective detection of BPA. Figure 7B illustrates a magnetic bead-based aptasensor-based quantum dot (QD) and its fluorescence detection of trace DEHP. Herein, in combination with a reduced graphene oxide (rGO)-based screening technique, a novel 60-mer aptamer has been successfully developed using systemic evolution of ligands by exponential enrichment (SELEX) technology. This aptamer-based sensor shows not only good selectivity of DEHP in the presence of other phthalate analogs but also excellent sensitivity as low as 5 pg/mL. As shown in Figure 7B–a, if DEHP is absent, the complex of a specific aptamer-coated magnetic bead conjugated with QD_565_ mainly binds to a DNA probe containing QD_655_ and its fluorescence is normalized QD_565_. As shown in Figure 7B–b in the presence of DEHP, DNA probes were found to be dissociated from a specific 60-mer aptamer leading to a decrease of normalized QD_655_/QD_565_ fluorescence ratios. Figure 7C demonstrates a facile detection of BPA using a combination between a specific aptasensor with SPR of AuNP-activated ZnO nanopencils. Due to many hot electrons from excited AuNPs, Au/ZnO nanocomposites exhibited an increment of photoelectrochemical signals higher than those from initial ZnO nanopencils. Based on a specific aptasensor, this sensor has good selectivity with excellent sensitivity of a detection limit as low as 0.5 nmol/L for BPA in real samples, as well as drinking water and milk.

As shown in Figure 8, a simple detection of trace DEHP has been developed using an aptamer and a Raman reporter of 5,5′-dithio-bis(2-nitrobenzoic acid) (DTNB) molecule [91]. Since AgNP surfaces have been functionalized with DTNB, NaCl-induced aggregations of AgNPs with a silica shell can become an efficient SERS system. These AgNPs–SiO_2_ structures have been functionalized with (3-aminopropyl) trimethoxy silane (APTMS) to form a primary amine layer on the surface. Subsequently, the surface of AgNPs–SiO_2_–NH_2_ structures has been immobilized with DEHP–COOH molecules by an amide binding through *N*-ethyl-*N*′-(3-dimethylaminopropyl) carbodiimide hydrochloride (EDC) and *N*-hydroxysulfosuccinimide (NHS) as a crosslinking reaction. DEHP aptamers have been immobilized on the surface of magnetic particles. There is a competitive binding between DEHP molecules and SERS silica particles with aptamer-coated magnetic particles. DEHP concentration has been quantitatively determined by comparing the signals of Raman reporter on SERS silica particles. Since DEHP has been widely used in plastic products, aptasensor-assisted SERS sensors can achieve highly selective detection of DEHP, despite interferents with analogous structures, which may provide numerous applications in food and environmental analysis.

As summarized in Table 1, in the past decades, numerous novel optical sensors based on plasmonic resonance have emerged for indoor SVOC detection with improving aspects such as economical and reproducible benefits, as well as selectivity and sensitivity. The following points have been emphasized and evaluated on the current optical sensor methods. There are some perspectives that will be addressed in the near future such as: (1) continuing to improve optical sensor methods based on not only pure noble metallic substances but also hybrid nanostructured materials, (2) developing more aptasensors that exhibit highly selective specific sensors, (3) several novel nano-systems should be developed that can demonstrate the capability of multiple detections of analytically equivalent molecules under complex conditions, and (4) discovering a potential nanosystem used for simultaneous detection of both SERS and fluorescence.

## 5. Conclusions

Plasmonic resonance-based optical sensors have emerged as a powerful analytical tool that assists in optical detection of SVOC emission into environments. The effects of indoor SVOC exposure on human health through many pathways of dermal, inhalation, and non-dietary and dietary ingestion have been investigated in previous reports [12,13,55], and it still requires further studies using functional plasmonic nanomaterials and related detection methodologies. SVOCs as chemical additives have been widely used in many household products, inducing SVOC emission from resources to indoor air.

Effective sensing of indoor SVOCs using optical sensor methods require not only facile detection but also selective and rapid identification to warn of the early risks of the impact on human health. Both the novel nanoplatforms and aptasensors have been introduced to efficient optical sensors, including SERS, localized surface plasmon, and colorimetric sensor methodologies in a real indoor environment. Potential applications of plasmonic nanoplatforms and novel aptasensors have been discussed for optical detection of indoor SVOCs from surface-functionalized materials. Extensive studies of indoor SVOC detection are needed to maintain a clean, safe, and healthy environment. Functional plasmonic materials should be critical in optical detection of SVOCs in the future.

## Figures and Tables

**Figure 1 nanomaterials-11-02619-f001:**
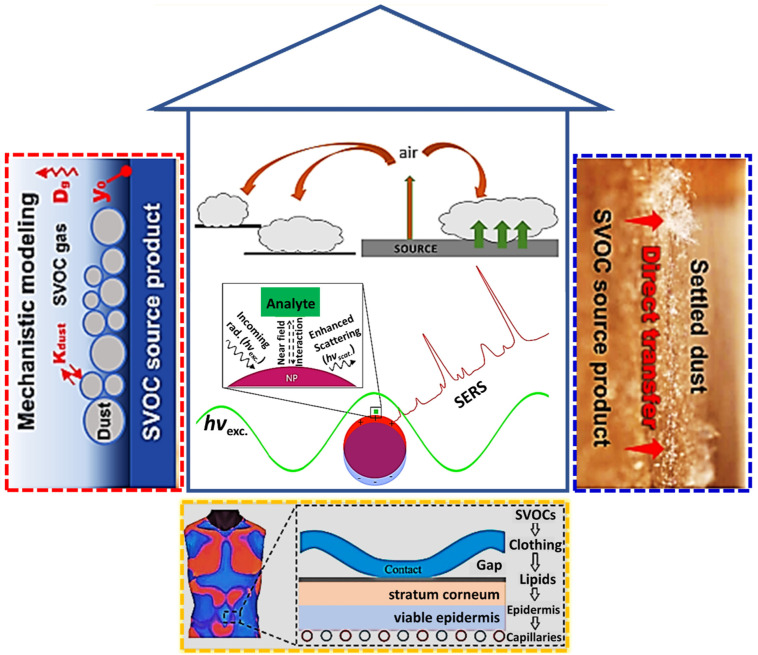
Applications of plasmonic resonance-based SERS sensors for on-site detection of indoor SVOCs. Adapted from [9,18,22,32].

**Figure 2 nanomaterials-11-02619-f002:**
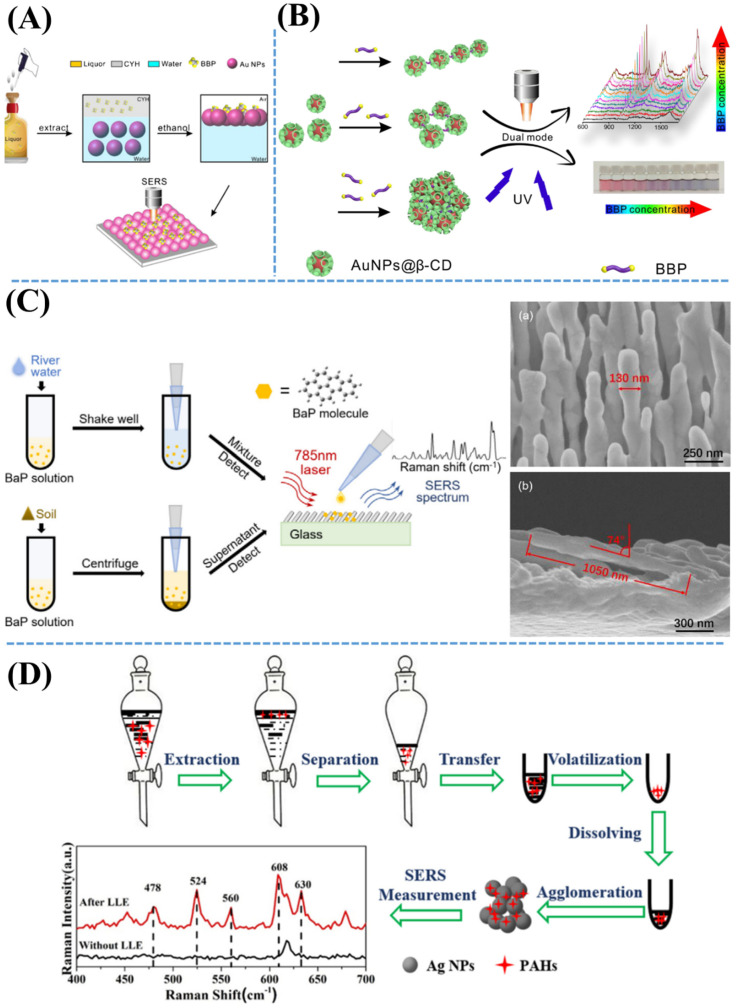
(**A**) Schematic illustration of the self-assembly of AuNPs arrays-introduced SERS detection of BBP after a process of molecule extraction from liquor by CYH. Adapted from [65]. (**B**) Diagram of BBP detection using β-cyclodextrin-stabilized AuNPs by means of either SERS spectra or colorimetric sensing. Adapted from [66]. (**C**) Demonstration of the silver nanorod (AgNR) substrate-based SERS detection for BaP. Inserted SEM images show (**C**–**a**) top-view and (**C**–**b**) cross-sectional view of AgNR substrate. Adapted from [67]. (**D**) Diagram of trace detection of polycyclic aromatic hydrocarbons (PAHs) using AgNP-assisted SERS technology. Adapted from [61].

**Figure 3 nanomaterials-11-02619-f003:**
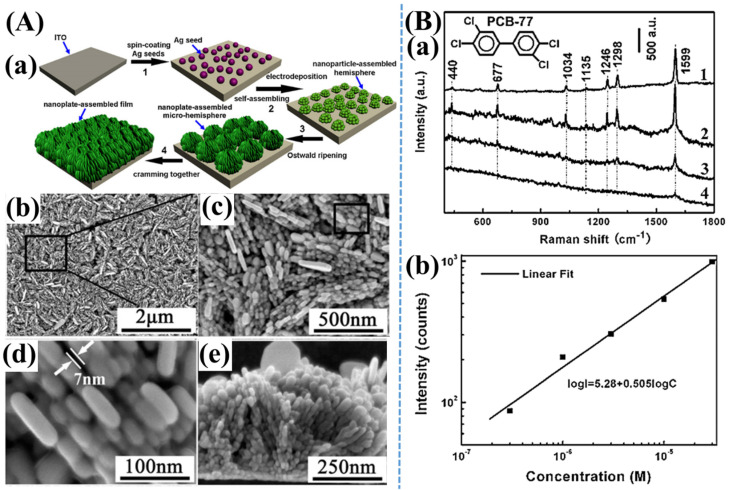
(**A**–**a**) Schematic diagram of sensitive SERS detection of 3,3′,4,4′-tetrachlorobiphenyl (PCB-77) based on the fabrication of vertically-aligned silver nanoplates assembled on indium tin oxide (ITO) substrates. SEM images of the Ag nanoplate-based SERS substrate at different scales including (**A**–**b**) 2 µm and (**A**–**c**) 500 nm. (**A**–**d**) SEM image shows a magnified view of the black color rectangular region marked in (**A**–**c**). (**A**–**e**) Side-view SEM image of the Ag nanoplate-based SERS platform. (**B**–**a**) Raman spectra of (1) pure PCB-77 powder, SERS spectra of PCB-77 with different concentrations of (2) 10^−4^ M, (3) 3 × 10^−6^ M, and (4) 3 × 10^−7^ M, respectively, using Ag nanoplate-assembled substrates. All Raman spectra were acquired with an acquisition time of 60 s. (**B**–**b**) Using Ag nanoplate-based SERS platform, a linear curve has been obtained for the logarithmic values for the intensities at 1599 cm^−1^ correlated with the concentrations of PCB-77. Adapted from [47].

**Figure 4 nanomaterials-11-02619-f004:**
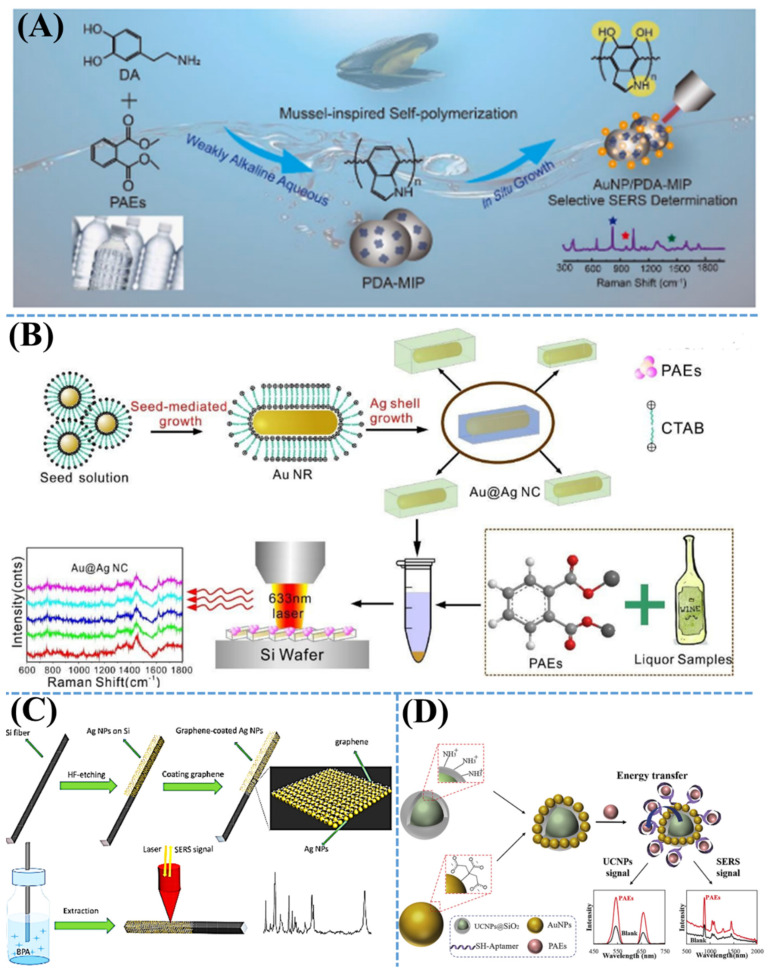
(**A**) Schematic illustration of SERS substrate for selective detection of PAEs using AuNP-coated PDA-MIP template. Adapted from [74]. (**B**) Diagram of SERS sensitive detection of PAEs using bimetallic plasmonic Au@Ag nanocuboids. Adapted from [73]. (**C**) Demonstration of SERS-active solid phase microextraction fiber detection of BPA. Adapted from [72]. (**D**) Diagram of SERS and fluorescence detection of PAEs using aptamer-assisted AuNP hybrid nanostructures. Adapted from [71].

**Figure 5 nanomaterials-11-02619-f005:**
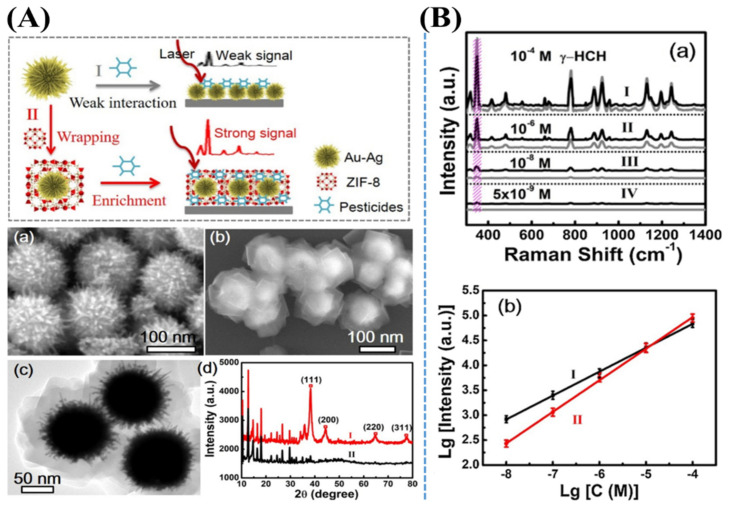
Schematic illustration of SERS detection of hexachlorocyclohexane pesticides based on urchin-like Au-AgNPs. (**A**–**a**) Scanning electron microscope (SEM) image shows the structures of urchin-like Au-AgNPs. (**A**–**b**) SEM image and (**A**–**c**) transmission electron microscope (TEM) image of the nanomaterial products obtained after adding the ZIF-8-wrapped urchin-like Au-AgNPs into the precursor solution and heating them at 40 °C for 3 h. (**A**–**d**) X-ray diffraction patterns including (I) nanomaterial products and (II) pure ZIF-8 achieved by heating the precursor solution at 40 °C for 3 h, respectively. (**B**–**a**) SERS spectra of γ-HCH on the ZIF-8 wrapped urchin-like Au-AgNPs (black), and bare urchin-like Au-AgNPs (grey) after soaking in γ-HCH solutions with different concentrations such as (I) 1 × 10^−4^ M, (II) 1 × 10^−6^ M, (III) 1 × 10^−8^ M, and (IV) 5 × 10^−9^ M, respectively, for both samples. The shadowed regions indicated that the characteristic peak of γ-HCH at 345 cm^−1^. (**B**–**b**) Linear curves of logarithmic plots of γ-HCH concentrations correlated to the Raman peak intensities at ca. 345 cm^−1^ for the ZIF-8 wrapped (I), and (II) bare urchin-like Au-AgNPs. Adapted from [40].

**Figure 6 nanomaterials-11-02619-f006:**
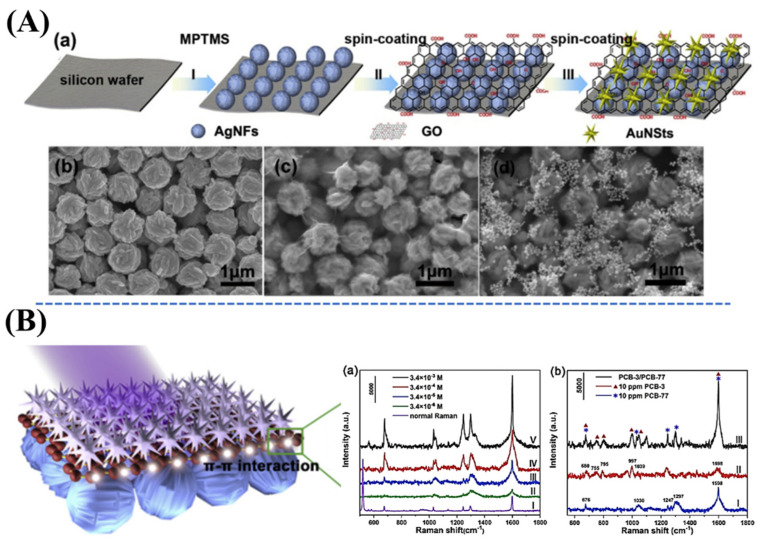
(**A**–**a**) Schematic demonstration of the fabrication process for a novel hybrid nanomaterial as well as Au nanostars–GO–flower-inspired Ag nanoparticles (AuNSts-GO-AgNFs) for ultra-sensitive SERS detection of PCBs. SEM images of (**A**–**b**) AgNFs, (**A**–**c**) AgNFs-GO, and (**A**–**d**) AgNFs-GO-AuNSts. (**B**–**a**) Normal Raman spectrum and SERS spectra of PCB-77 on this substrate with different concentrations. (**B**–**b**) Compared SERS spectroscopy of PCB-77, PCB-3, and PCB-3/PCB-77 mixture. Adapted from [38].

**Figure 7 nanomaterials-11-02619-f007:**
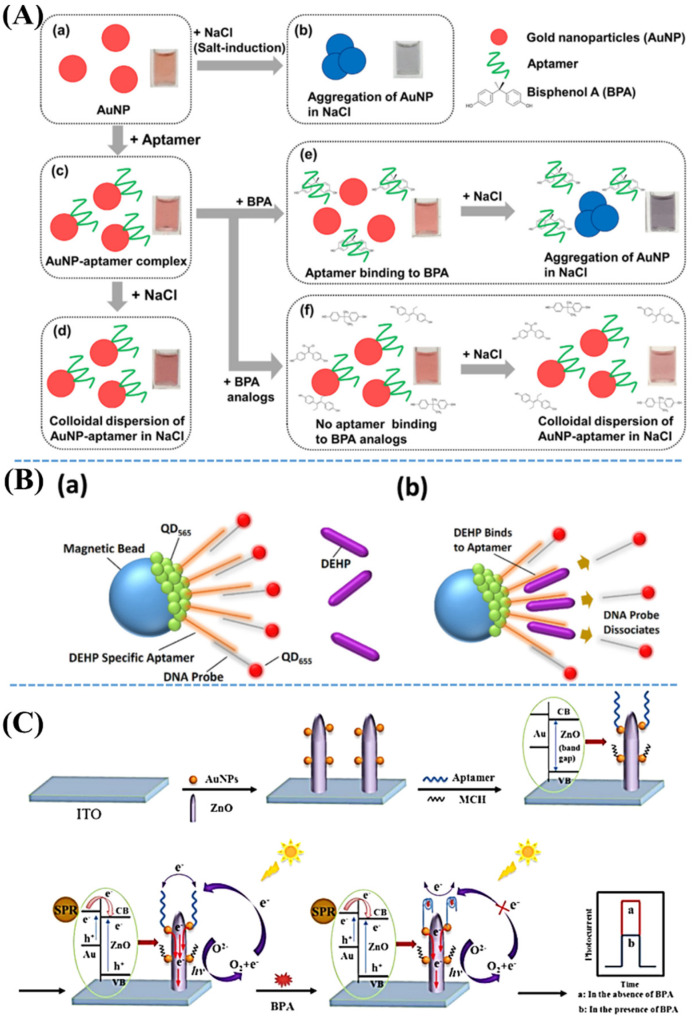
(**A**) Schematic illustration of colorimetric BPA detection based on an aptasensor using AuNPs. (**A**–**a**) AuNP solutions. (**A**–**b**) NaCl-induced AuNP aggregations. (**A**–**c**) aptamer addition leading to form AuNP-aptamer. (**A**–**d**) AuNP-aptamer is still well-dispersed after adding NaCl. (**A**–**e**) BPA presence inducing the binding between aptamer and BPA. Subsequent addition of NaCl resulting in the aggregation of AuNP-aptamer complexes. (**A**–**f**) Presence of BPA analogs leading to low binding affinity. Despite the addition of NaCl, AuNP-aptamer complexes are still dispersing well. Adapted from [82]. Schematic illustration of the quantum dot aptasensors (**B**–**a**) in the absence and (**B**–**b**) in the presence of DEHP. Adapted from [83]. (**C**) Diagram of the fabricated photoelectrochemical aptasensors using AuNPs and ZnO nanopencils functionalized with BPA aptamers, which was aimed for BPA detection by means of the SPR technique. Adapted from [85].

**Figure 8 nanomaterials-11-02619-f008:**
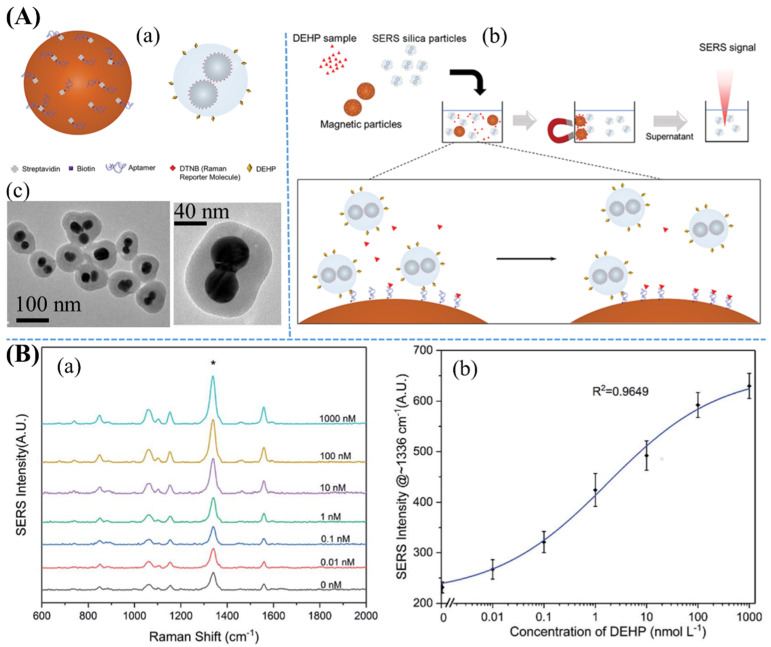
(**A**–**a**) Left-hand side showing the surface of magnetic particles was functionalized by components such as streptavidin, biotin, and DEHP aptamer, whereas right-hand side shows silica-capped Ag nanoclusters, and the surface of silica shell was functionalized with DTNB and conjugated with DEHP. (**A**–**b**) SERS detection of DEHP sample based on competitive binding reaction between silica particles and DEHP molecules with magnetic particles. (**A**–**c**) TEM images of silica particle-capped Ag nanocluster with scale bar at 100 and 40 nm, respectively. (**B**–**a**) SERS spectra of Raman reporter correlated with different concentrations of DEHP in 0.1 M phosphate-buffered saline (pH 7.4), which exhibited a behavior response of the aptasensor. (**B**–**b**) DEHP concentration has responded to SERS intensity of the peak at 1336 cm^−1^. Error bars represent standard deviation. Adapted from [91].

**Table 1 nanomaterials-11-02619-t001:** Performance of plasmonic resonance-based optical sensors for detection of indoor SVOCs.

Plasmonic Structures	Optical Sensors	SVOCs	Limit of Detection	Reference
Ag and Au probes	SERS	2-naphthalenethiol	0.1 ppb	[34]
Ultrathin tin oxide layer-wrapped AuNPs	SERS	phenyl phosphonic acid		[35]
GO-anisotropic noble metal hybrid systems	SERS	PCBs	3.4 × 10^−6^ M	[38]
Urchin-like Au-Ag crystals	SERS	HCH pesticides	>1.5 ppb	[40]
AuNPs	SERS	BZT		[44]
Ag nanoplate-assembled film	SERS	3,3′,4,4′-tetrachlorobiphenyl	10^−6^ M	[47]
AgNPs	SERS	16 typical polycyclic aromatic hydrocarbons	100–0.1 μg/L	[61]
AuNPs	SERS	BPA	0.1 ng/mL	[62]
AgNPs	SERS	BPA	5 × 10^−8^ M	[63]
AgNPs	SERS	bisphenol A, B, and S	10^−7^ M	[64]
AuNPs	SERS	BBP	1.3 mg/kg	[65]
AuNPs	SERS	BBP	0.01 μM	[66]
Ag nanorods	SERS	BaP	1 ppm	[67]
Au nanospheres@Ag nanocubes	SERS	BBP	10^−9^ M	[69]
Au@AgNPs	SERS	BPA	2.8 pg/mL	[70]
AuNP-decorated up-conversion nanoparticles	SERS	dibutyl phthalate	0.0108 ng/mL	[71]
Graphene monolayer-coated AgNPs	SERS	BPA	1 μg/L	[72]
Bimetallic plasmonic Au@Ag nanocuboids	SERS	phthalate plasticizers	10^−9^ M	[73]
AuNP-coated MIP template	SERS	phthalate plasticizers	10^−10^ M	[74]
Au nano-antennae fabricated optical fibers	Coupled localized SPR	BPA	330 ± 70 aM	[80]
Asymmetric plasmonic aptasensors	UV-Vis	BPA	0.008 ng/mL	[81]
AuNPs-based colorimetric aptasensors	Colorimetric	BPA	1 pg/mL	[82]
AuNPs-sensitized ZnO	SPR	BPA	0.5 nmol/L	[85]
Silica-coated Ag nanoclusters	SERS	DEHP	8 pM	[91]
Gold layer-coated SiO_2_ nanostructured pillars	SERS	benzotriazole	17.6 µg/L	[92]
AuNPs	SERS	pyrene	0.4 nM	[93]
		anthracene	4.4 nM	
Au@Ag@ hexakisphosphate /1-dodecanethiol	SERS	diethylhexyl phthalate	10^−8^ M	[94]
Ag/SiO_2_	SERS	DEHP	100 ppm	[95]
		BBP		
		dibutyl phthalate		
Au nanostructures	SERS	BaP	0.026 mg/L	[96]
		fluoranthene	0.064 mg/L	
		naphthalene	3.94 mg/L	
Three-dimensional SERS substrates based on porous material and pH 13 AuNPs	SERS	phenanthrene	8.3 × 10^−10^ M	[97]
		pyrene	2.1 × 10^−10^ M	
		BaP	3.8 × 10^−10^ M	
		benzo(k)fluoranthene	1.7 × 10^−10^ M	
Thioctic acid-modified Ag nanoplates on Cu foils		fluoranthene	0.1 ng/mL–0.1 mg/mL	[98]
Alkanethiol-Ag(Au)	Raman	polybrominated diphenyl ethers	1.2 × 10^2^ μg/L	[99]
Ag colloids	SERS	naphthalene	10^−12^ M	[100]
		phenanthrene	10^−10^ M	
Bowl-shaped Ag	SERS	anthracene	8 nM	[101]
		pyrene	40 nM	
Au-colloid substrates	SERS	naphthalene	1.38 μg/L	[102]
		phenanthrene	0.23 μg/L	
		pyrene	0.45 μg/L	

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
