# Peer review of "Surface-Enhanced Raman Sensing of Semi-Volatile Organic Compounds by Plasmonic Nanostructures"

_nanomaterials, 2021, doi:10.3390/nano11102619_

Round 1

Reviewer 1 Report

This review is seriously lacking in the text that would teach people something about this field. The figures are put together as a collection of figures from previous works but they are not described very much at all in the text and are too difficult to get any information from on their own. The authors didn't even bother editing the images enough and there are multiple a), b) or A, B indicators in Figures like 3. It looks like the authors did not put much effort in at all for this manuscript.

Author Response

Reply to Reviewer #1's Comments

The authors deeply appreciate the Reviewer #1 for giving helpful comments.

Comments and Suggestions for Authors: This review is seriously lacking in the text that would teach people something about this field. The figures are put together as a collection of figures from previous works but they are not described very much at all in the text and are too difficult to get any information from on their own. The authors didn't even bother editing the images enough and there are multiple a), b) or A, B indicators in Figures like 3. It looks like the authors did not put much effort in at all for this manuscript.

→ According to Reviewer #1’s comments, the manuscript is revised as follows.

1) We described more detail about Figure 2 at the 120-141st lines of the manuscript as follows:

“Figure 2A introduces an efficient SERS detection of BBP using ~18 nm spherical AuNPs, first BBPs have been isolated by extracting with organic solvent as cyclohexane (CYH). Subsequently, an amount of ethanol has been added to this mixture inducing to self-assemble AuNPs at the interface between CYH and water. Within 10–20 s, aggregated AuNP arrays have been formed with many hotspots. After being transferred onto the sur-face of clean silicon wafers, self-assembled AuNP arrays containing BBPs have been checked by means of SERS. Figure 2B shows a schematic diagram of an efficient colori-metric and SERS sensing of BBP d using β-cyclodextrin (β-CD) stabilized AuNPs. β-CD molecules have played significant roles of Raman probes to capture BBP. Due to the AuNP-correlated hot spots, BBP could be quantified either with the naked eye or by means of SERS. Figure 2C demonstrates a silver nanorod-based SERS method for a trace detection of BaP in real samples as well as river water and soil. Due to the unique structures of SERS substrates with highly dense hotspots, the detection limit of BaP has been estimated to be as low as 1 ppm and 10 ppm in river water and soil, respectively. Figure 2D illustrates a schematic diagram for AgNP-assisted SERS detection of 16 typical PAHs in water. Based on plasmonic AgNPs and the liquid extraction step, the SERS method has been success-fully applied to a trace detection of PAHs in real water environments containing contam-inants. Herein, PAHs have been isolated from water by extraction with a nonpolar organic solvent. Subsequently, three steps including separation, transfer, and volatilization have been performed. Finally, acetonitrile has been used to elute the extracted PAHs to obtain solution for measured SERS spectra.”

2) We provided a new Figure 3 which has been edited as follows.

Figure 3. (A-a) Schematic diagram of SERS sensitive detection of 3,3’,4,4’-tetrachlorobiphenyl (PCB-77) based on the fabrication of well-aligned vertically silver nanoplate-assembled on indium tin oxide (ITO) substrate. Inserted SEM images of the Ag nanoplate-based SERS substrate at different scale bar including (A-b) 2 µm, and (A-c) 500 nm. Also inserted SEM image (A-d) shows close-up view of the black color rectangular region marked in (A-c). Another inserted SEM (A-e) side view of the Ag nanoplate-based SERS platform. (B-a) Raman spectra of (1) pure PCB-77 powder, SERS spectra of PCB-77 with different concentrations using Ag nanoplate-assembled substrate including (2) 10−4 M, (3) 3 × 10−6 M, and (4) 3 × 10−7 M, respectively. All Raman spectra were acquired at 60 s with acquisition time. (B-b) Using Ag nanoplate-based SERS platform, a linear curve of logarithmic intensities (1599 cm−1)-correlated the concentrations of PCB-77 has been performed. Adapted from ref. [47].

3) We described more detail about Figure 4 at the 202-229th lines of the manuscript as follows.

“As shown in Figure 4, several studies on hybrid nanomaterial-based SERS methods have been reported on a trace detection of various indoor SVOCs such as PAEs and BPA. For example, based on the self-oxidative-polymerization of dopamine, the bio-inspired nanostructures of molecularly imprinted polymers (MIP) have been successfully designed as a template for in situ fabrication of AuNPs [74]. AuNP-coated MIP template has been a three-dimensional (3D) nanoplatform, which has been used for selective and sensitive de-tection of phthalate plasticizers through SERS spectroscopy. These SERS plasmonic nanostructures have been well controlled to produce many “hot spots” on the surface of nanocomposites. Figure 4B illustrates a SERS detection of PAEs in liquid samples based on plasmonic core-shell nano-cuboid of bimetallic Au@Ag. Herein, Au@Ag nano-cuboids have been successfully fabricated by Au nanorod core and Ag cuboid shell leading to in-duce high more plasmonic resonance than pristine Au nanorods. Due to a unique struc-ture of Au@Ag nanocomposites to induce strong signals, this SERS sensor indicated that detection limits of BBP and DEHP were estimated to be as low as 10-9 M. As shown in Fig-ure 4C, a sensitive detection of BPA has been investigated by a solid-phase microextrac-tion (SPME) and the SERS method. AgNPs were in situ synthesized on the surface of Si fi-ber leading to form AgNP-coated Si fiber. Subsequently, this structure has been modified with a monolayer of graphene resulting to generate a novel nanocomposite including Si fiber, AgNPs, and graphene. This unique structure exhibits not only as a SERS platform but also as an SPME fiber to detect BPA. Due to the effective cooperation of SPME and SERS, this technique shows an excellent capability of BPA detection with a detection limit as low as 1 μg/L. Figure 4D demonstrates a PAE detection by means of SERS up-conversion fluorescence. Using AuNPs to decorate up-conversion nanoparticles (UCNPs), a new bimodal platform has been successfully developed in this report. In this nanocomposite, UCNPs have been employed as a fluorescent signal molecule, whereas AuNPs have been used as SERS templates with an aptamer served as a targeted detection of PAEs. This nanocomposites-assisted detection of PAEs has demonstrated the detection limes of 0.0087 ng/mL and 0.0108 ng/mL for fluorescence sensors and SERS methods, re-spectively.”

4) We described more detail about Figure 7 at the 304-329th lines of the manuscript as follows.

“Figure 7A shows an aptasensor-based AuNP-assisted colorimetric detection of BPA. Herein, the aptasensor has been well-designed to consist of AuNPs and a specific 24-bp aptamer of BPA. Aptamer-conjugated AuNP complex has been formed via an electrostatic interaction. This complex will induce a SPR shift in the presence of BPA. Pristine AuNPs exhibits a red color. Because of the presence of BPA leading to the aggregation of AuNPs, the color may change to blue, which can be observe with the naked eye. Based on [BPA]-correlated blue color changes of aptamer-conjugated AuNP complex, the colorimet-ric detection has been successfully performed with a limit as low as 0.004 nM. Despite the presence of various BPA analogs, this method achieved a selective detection of BPA. Fig-ure 7B illustrates a magnetic bead-based aptasensor-based quantum dot (QD) and its flu-orescence detection of trace DEHP. Herein, in combination with a reduced graphene oxide (rGO)-based screening technique, a novel 60-mer aptamer has been successfully devel-oped using systemic evolution of ligands by exponential enrichment (SELEX) technology. This aptamer-based sensor shows that not only good targeted selectivity of DEHP in the presence of other phthalate analogs but also excellent sensitivity as low as 5 pg/mL. As shown in Figure 7(B-a), if DEHP is absent, the complex of specific aptamer-coated mag-netic bead conjugated with QD565 mainly binds to DNA probe containing QD655 and its fluorescence is normalized QD565. As shown in Figure 7(B-b) in the presence of DEHP, DNA probes were found to be dissociated from specific 60-mer aptamer leading to a de-crease of normalized QD655/QD565 fluorescence ratios. Figure 7C demonstrates a facile de-tection of BPA using a combination between a specific aptasensor with SPR of AuNP-activated ZnO nano-pencils. Due to many hot electrons from excited AuNPs, Au/ZnO nanocomposites exhibited increment of photoelectrochemical signals higher than that from initial ZnO nanopencils. Based on a specific aptasensor, this sensor has a good selectivity with an excellent sensitivity of a detection limit as low as 0.5 nmol/L of BPA in real samples as well as drinking water and milk.”

5) We described more detail about Figure 8 at the 340-355th lines of the manuscript as follows.

“As shown in Figure 8, a simple detection of trace DEHP has been developed using an aptamer and a Raman reporter of 5,5’-dithio-bis(2-nitrobenzoic acid (DTNB) molecule [91]. Since AgNP surfaces have been functionalized with DTNB, NaCl-induced aggregations of AgNPs with a silica shell can become an efficient SERS system. These AgNPs–SiO2 struc-tures have been functionalized with (3-aminopropyl) trimethoxy silane (APTMS) to form a primary amine layer on surface. Subsequently, the surface of AgNPs–SiO2–NH2 structures has been immobilized with DEHP–COOH molecules by an amide binding through N-ethyl-N’-(3-dimethylaminopropyl) carbodiimide hydrochloride (EDC) and N-hydroxysulfosuccinimide (NHS) as a crosslinking reaction. DEHP aptamers have been immobilized on the surface of magnetic particles. There is a competitive binding between DEHP molecules and SERS silica particles with aptamer-coated magnetic particles. DEHP concentration has been quantitatively determined by comparing the signals of Raman re-porter on SERS silica particles. Since DEHP has been widely used in plastic products, ap-tasensor-assisted SERS sensors can achieve highly selective detection of DEHP, despite interferents with analogous structures, which may provide numerous applications in food and environmental analysis.”

Finally, the authors would like to thank Reviewer #1 for giving valuable comments. Thank you.

Reviewer 2 Report

The review is devoted to studies in the field of detection of volatiles using surface-enhanced Raman spectroscopy (SERS). The manuscript contains a lot of useful information about the latest advances in this scientific area. The article can be recommended for publication after minor improvements.

  1. It would be great to insert additional references into the introduction which are dedicated to gas detection using SERS, for example

Wang J. et al. // Anal. Chem. 2011. V. 83, â„– 6. P. 2243;

Wong C.L. et al. // Anal. Chim. Acta. 2014. V. 844. P. 54;

 Petrov D.V. et al // Opt. Lett. 2017. V. 42, â„– 22. P. 4728,

Sharma S.K. et al. // ChemistrySelect. 2017. V. 2, â„– 24. P. 6961.

Oh M. et al. // J. Raman Spectrosc. 2018. V. 49, â„– 5. P. 800.

  1. Figure 1. It is necessary to give full names instead of abbreviations like «SC» and «VE».

Author Response

Reply to Reviewer #2's Comments

The authors deeply appreciate the Reviewer #2 for giving helpful comments.

Comments and Suggestions for Authors: The review is devoted to studies in the field of detection of volatiles using surface-enhanced Raman spectroscopy (SERS). The manuscript contains a lot of useful information about the latest advances in this scientific area. The article can be recommended for publication after minor improvements.

1) It would be great to insert additional references into the introduction which are dedicated to gas detection using SERS, for example

Wang J. et al. // Anal. Chem. 2011. V. 83, â„– 6. P. 2243;

Wong C.L. et al. // Anal. Chim. Acta. 2014. V. 844. P. 54;

Petrov D.V. et al // Opt. Lett. 2017. V. 42, â„– 22. P. 4728,

Sharma S.K. et al. // ChemistrySelect. 2017. V. 2, â„– 24. P. 6961.

Oh M. et al. // J. Raman Spectrosc. 2018. V. 49, â„– 5. P. 800.

→ In the revised manuscript, we added two new recently published references based on the latest year published among the suggested ones.

  1. Oh, M.-K.; De, R.; Yim, S.-Y., Highly sensitive VOC gas sensor employing deep cooling of SERS film. J. Raman Spectrosc. 2018, 49, 800-809.
  2. Petrov, D. V.; Zaripov, A. R.; Toropov, N. A., Enhancement of Raman scattering of a gaseous medium near the surface of a silver holographic grating. Opt. Lett. 2017, 42, 4728-4731.

2) Figure 1. It is necessary to give full names instead of abbreviations like «SC» and «VE».

→ In Figure 1, we have provided full names of “stratum corneum” and “viable epidermis” for “SV” and “VE”, respectively.

Finally, the authors would like to thank Reviewer #2 for giving valuable comments. Thank you.

Reviewer 3 Report

It is very nice review article, in which Ly et al. described applications of surface-enhanced Raman spectroscopy (SERS) for sensing of semi-volatile organic compounds. I have read this review article with great pleasure and, in my opinion, it is possible to publish this work without any corrections in its current form. Perhaps it is only worth considering including a list of used abbreviations with their explanations at the end of the paper (this possible improvement can be, however, introduced at the proofreading stage).

Author Response

Reply to Reviewer #3's Comments

The authors deeply appreciate the Reviewer #3 for giving helpful comments.

Comments and Suggestions for Authors: It is very nice review article, in which Ly et al. described applications of surface-enhanced Raman spectroscopy (SERS) for sensing of semi-volatile organic compounds. I have read this review article with great pleasure and, in my opinion, it is possible to publish this work without any corrections in its current form. Perhaps it is only worth considering including a list of used abbreviations with their explanations at the end of the paper (this possible improvement can be, however, introduced at the proofreading stage).

→ According to Reviewer #3’s comments, a list of abbreviations has been added at the end of the text in the revised manuscript,.

List of Abbreviations:

APTMS                        (3-aminopropyl) trimethoxy silane

BaP                                 benzo(a)pyrene

BZT                                benzothiazole

BPA                               bisphenol A

BBP                                butyl benzyl phthalate

β-CD                             β-cyclodextrin

CYH                              cyclohexane

DEHP                            di-2-ethylhexyl phthalate

DNA                               deoxyribonucleic acid

DTNB                             5,5’-dithio-bis(2-nitrobenzoic acid

EDC                                N-ethyl-N’-(3-dimethylaminopropyl) carbodiimide hydrochloride

GO                                  graphene oxide

HCH                              hexachlorocyclohexane

MIP                                 molecularly imprinted polymers

NHS                                N-hydroxysulfosuccinimide

NP                                   Nanoparticle

NSt                                  Nanostar

PAE                                phthalic acid ester

PAH                                polycyclic aromatic hydrocarbon

PCB                                polychlorinated biphenyl

QD                                quantum dot

rGO                                 reduced graphene oxide

RNA                              ribonucleic acid

SELEX                        systemic evolution of ligands by exponential enrichment

SEM                              scanning electron microscope

SERS                              surface-enhanced Raman scattering

SPME                             solid-phase microextraction

SPR                                 surface plasmon resonance

SVOC                             semi-volatile organic compounds

TEM                              transmission electron microscope

UCNP                             up-conversion nanoparticle

ZIF                                zeolite imidazole framework

Finally, the authors would like to thank Reviewer #3 for giving positive comments. Thank you.

Round 2

Reviewer 1 Report

The authors addresses a lot of the original concerns by describing things more in the text. This seems to be a collection of data put into a review and still could be more accessible to readers but it looks much more sound now.